# Profiling Cancer Cells by Cell-SELEX: Use of Aptamers for Discovery of Actionable Biomarkers and Therapeutic Applications Thereof

**DOI:** 10.3390/pharmaceutics14010028

**Published:** 2021-12-24

**Authors:** Sarah Shigdar, Lisa Agnello, Monica Fedele, Simona Camorani, Laura Cerchia

**Affiliations:** 1School of Medicine, Deakin University, Geelong 3220, Australia; sarah.shigdar@deakin.edu.au; 2Institute for Mental and Physical Health and Clinical Translation, School of Medicine, Deakin University, Geelong 3220, Australia; 3Institute of Experimental Endocrinology and Oncology “Gaetano Salvatore”, CNR, Via S. Pansini 5, 80131 Naples, Italy; lisa.agnello@ieos.cnr.it (L.A.); mfedele@unina.it (M.F.); simona.camorani@ieos.cnr.it (S.C.); 4Department of Precision Medicine, University of Campania “L. Vanvitelli”, S. Andrea Delle Dame-Via L. De Crecchio 7, 80138 Naples, Italy

**Keywords:** cell-SELEX, aptamer, cancer cell phenotype, biomarker discovery, cell-profiling, targeted therapy, aptamer-antibody conjugates, aptamer guided-nanomedicines, targeted delivery

## Abstract

The identification of tumor cell-specific surface markers is a key step towards personalized cancer medicine, allowing early assessment and accurate diagnosis, and development of efficacious targeted therapies. Despite significant efforts, currently the spectrum of cell membrane targets associated with approved treatments is still limited, causing an inability to treat a large number of cancers. What mainly limits the number of ideal clinical biomarkers is the high complexity and heterogeneity of several human cancers and still-limited methods for molecular profiling of specific cancer types. Thanks to the simplicity, versatility and effectiveness of its application, cell-SELEX (Systematic Evolution of Ligands by Exponential Enrichment) technology is a valid complement to the present strategies for biomarkers’ discovery. We and other researchers worldwide are attempting to apply cell-SELEX to the generation of oligonucleotide aptamers as tools for both identifying new cancer biomarkers and targeting them by innovative therapeutic strategies. In this review, we discuss the potential of cell-SELEX for increasing the currently limited repertoire of actionable cancer cell-surface biomarkers and focus on the use of the selected aptamers as components of innovative conjugates and nano-formulations for cancer therapy.

## 1. Introduction

Cancer is the second leading cause of death worldwide, and its incidence is estimated to increase in the next decades, with over 20 million new cancer cases annually expected by 2025 [1,2]. Currently, chemotherapy and radiotherapy remain the main treatment options for the majority of cancers, but in killing highly proliferative cells they are unable to effectively distinguish tumor cells from rapidly replicating non-malignant cells, thus producing severe toxic side effects [3,4]. Therefore, continuous efforts are dedicated to developing innovative tumor-specific approaches in order to overcome the non-specificity of conventional therapies.

Identifying the molecular differences between healthy and tumor cells and among different tumor types and subtypes is the key to developing therapeutic strategies that specifically target the molecular alterations implicated in cancer. To date, a wide range of cancer-specific biomarkers and their complex combinations have been identified that specifically reflect the phenotype of diseased cells and inform about pathological states, thus becoming significant players in clinical diagnosis and therapy [5].

The cancer cell-surface proteome, or surfaceome, which is the set of proteins expressed on cancer cells, represents the main source for the selection of relevant biomarkers for prognostic, diagnostic and therapeutic applications [6]. Indeed, several cell-surface proteins, often found mutated or abnormally expressed on cancer cells, play essential roles in the interaction with the extracellular tumor environment (TME), in cell–cell communication, in nutrient homeostasis, in cell signaling, growth, proliferation and spread, and in defining the immunologic identity of cancer cells.

In 1977, the United States food and drug administration (FDA) approved Tamoxifen, a selective estrogen receptor (ER) modulator, for the treatment of patients with ER-positive breast cancer. Later, the trastuzumab monoclonal antibody (mAb) and gefitinib tyrosine kinase inhibitor (TKI), were as well approved for the treatment of epidermal growth factor receptor 2 (HER2)-positive breast cancers and certain types of epidermal growth factor receptor (EGFR) mutated-expressing non-small-cell lung cancers (NSCLC), respectively. After that, different drugs were developed to manage some cell-surface protein-driven cancers [7]. However, despite the large number of cell-surface proteins encoded by the human genome and the therapeutic implication for targeting the cell-surface proteome, antibodies against only approximately 20 cell-surface targets have been approved for cancer therapy [8,9], thus underlining the need to find novel surface targets, and combination thereof, for improving targeted therapy. Currently, the lack of effective strategies for profiling cancer cell-surface and selecting actionable targets represents one of the main reasons associated with this still too small number of effective antibody-based target therapies [10,11]. In recent years, the cell-SELEX (Systematic Evolution of Ligands by Exponential Enrichment) technology for the differential selection of oligonucleotide aptamers against a specific cancer-cell type has become the selection technique for the discovery of cell-surface markers [12,13,14]. Indeed, it allows selection, at the same time, of a set of aptamers acting as highly efficacious recognition elements for functional surface signatures of target cells. Importantly, these aptamers may be used to identify cell-surface molecules whose role is still unexplored [15]. This fulfills the great challenge of simultaneously targeting multiple proteins whose alterations, in concert, define the pathological state of the cell and are thus more informative for biomarker discovery than the alteration of a single protein.

In the following paragraphs, by first touching on the potential of cell-SELEX for generating cancer therapeutics, we aim to highlight the power of cell-SELEX for molecular subtyping of cancer cells and exploring the complex heterogeneity of aggressive cancers that hinders the discovery of targeted therapies. Furthermore, the aptamers used to date for the discovery of actionable biomarkers are reviewed with a focus on aptamer-based delivery strategies through those biomarkers identified for anticancer therapy.

## 2. Cell-SELEX: Potential of Aptamers as Cancer Targeted Therapeutics

Aptamers are highly structured short single-stranded (ss) DNA or RNA which, because of their primary sequence, fold into complex three-dimensional (3D) structures that bind at high affinity to a specific target. They are screened from large oligonucleotide libraries for binding to a wide range of targets including metals, proteins, peptides, carbohydrates, small molecules, toxins, viruses and live cells [16].

Because of their complex shapes, aptamers, often termed “chemical antibodies”, recognize their target with affinity similar or higher than antibodies. Indeed aptamer-target complexes have dissociation-constant (K_D_) values in the low nanomolar to picomolar range [17]. Compared to antibodies, aptamers possess many advantages for cancer therapeutic applications, such as low size that allows high tissue penetration, physiological stability, no or low immunogenicity and minimized batch-to-batch variation [18,19,20,21]. Moreover, aptamers support many kinds of chemical modifications, which improve their activity, half-life, renal clearance and, importantly, allow sophisticated combination therapies by conjugating aptamers with other types of innovative therapeutic molecules or conventional drugs [22,23,24].

The SELEX method (Figure 1) for aptamer generation consists of iterative rounds that include the following steps: (1) incubation of the random oligonucleotide library with the target molecule; (2) separation of unbound sequences; (3) elution of bound sequences from target; (4) amplification of the pool of target-binding sequences. Up to 8–20 selection cycles (depending on the nature of the target) are commonly required to enrich the library of aptamers with high affinity to the target, even if the continuous improvement of the technology has recently allowed for reducing the number of cycles and the time required for the whole process [25,26,27,28]. The introduction of counter-selection steps into each SELEX round allows depletion of the library of sequences that bind to non-target molecules. At the end of the selection, individual sequences are identified by classical cloning of the enriched library and/or high throughput Next-Generation Sequencing (NGS) technology. Using bioinformatic tools, it is possible to identify target-specific candidates and structural elements indicative of potential binding sites [29,30,31]. Subsequently, the best candidate sequences are tested for their ability to bind specifically to the target molecule.

Optimization of the binding affinity and specificity of aptamers for the target molecules can be obtained by post-SELEX modification. Computational aptamer structure and aptamer/protein docking modeling can guide the truncation of the sequence to its minimal binding site and introduction of mutation or chemical modification to improve targeting performance of the aptamer [32]. Several factors affect the interaction of the aptamer to its target, including the reaction medium, pH and temperature, thus binding modes of aptamers might change in different reaction media [33].

Cell-SELEX applies to entire living cells as the target for selection, by using either cells expressing a pre-identified target protein or as discussed below, a specific cell type, with no prior knowledge of cell-surface marker proteins.

In the first approach, by altering positive selection on cells modified to overexpress a surface protein of interest and counter-selection steps on parental cells, aptamers are generated that specifically bind to the chosen target embedded in its natural environment. Thus, cell-SELEX might be more advantageous than protein-SELEX against the same target but in a purified form, an approach that could lead to aptamers that are non-functional in physiological conditions [34,35].

In some cases, cell-targeting aptamers are applied as stand-alone antagonistic agents since they interfere with the function of the protein target. For instance, the binding of an aptamer to a transmembrane receptor (i.e., a receptor tyrosine kinase, RTK) may block its interaction with the natural ligand, other cell-surface proteins or the TME, thus ultimately affecting cancer-cell function [36]. On the other hand, for those cell-targeting aptamers that bind to unique biomarkers of cancer cells but are not able to modulate the function of the target, targeted anticancer strategies have been developed upon aptamer conjugation to secondary reagents, where the aptamer is used for targeting and the other molecule to exert some therapeutic effect [37].

Notably, thanks to their easy derivatization, cell-targeting aptamers have been linked to other aptamers [38,39,40] or antibodies [41,42] to improve cancer treatment or have been used to drive immune cells to cancer cells for cell-specific immunotherapy [43].

Further, a subset of cell-targeting aptamers actively internalizes, usually via receptor-mediated endocytosis, thus having the potential to deliver therapeutic payloads specifically into cancer target cells, upon direct conjugation to the drug or to different drug-loaded nano-formulations [44]. One of these aptamers has shown the ability to both transcytose and endocytose through and into cells, respectively, and deliver a therapeutic payload. The bifunctional aptamer, TEPP, linked two separate aptamers that targeted the epithelial cell adhesion molecule (EpCAM) on epithelial cancer cells, and the transferrin receptor that is overexpressed on the blood brain barrier (BBB) [45]. The binding affinity of the transferrin receptor (TfR) aptamer increased from ~500 nM in its single form, to ~110 nM when linked to the EpCAM aptamer, a binding affinity that has been shown to be suitable for transcytosis through the endothelial cells of the BBB [38,45,46]. Intercalation of doxorubicin (Dox) into the stem region had limited effects on the binding affinity. The aptamer was shown to transcytose the BBB and be internalized into cancer cells within 75 min following a tail vein injection in a mouse model of brain metastasis from triple-negative breast cancer (TNBC) [39].

## 3. Profiling Cancer Cells by Cell-SELEX

One of the greatest advantages of cell-SELEX is the possibility to perform the selection against a specific cell type without the prior knowledge of protein targets present on its surface. In such a way, a panel of aptamers can be generated that specifically recognizes the surface signature of the target cells and, through it, can distinguish those cells from the cells chosen for the counter-selection. These aptamers are applied as useful bioreagents for active cancer targeting and, notably, can be used as bait for the identification of new protein targets, playing an important role in biomarker discovery (Figure 2).

Cell-SELEX has been applied to different tumor types, including highly aggressive and heterogeneous cancers that lack well-defined biomarkers for a targeted therapy, with the intent to discover new aptamers able to bind to surface proteins, which differ in expression level between healthy and unhealthy cells or among different cancer cell phenotypes.

The first group who realized the opportunity to dissect biological complex targets using SELEX technology was Morris and Jensen’s group in 1998. They used human red blood cell membrane preparations as a complex mixture of potential targets to select a set of cell-specific ssDNA aptamers with high affinity for different cell membrane proteins [47]. It was not long before there was an understanding that this approach could be also applied to decode the specific surface signature of different kinds of cancer cells.

Tan’s group applied the SELEX technology to living cells to explore membrane protein biomarkers. In their pioneering work, they screened a DNA library on cultured precursor T cell acute lymphoblastic leukemia (ALL) to generate a panel of aptamers specifically discriminating target cells from human Burkitt’s lymphoma cells, used in the counter-selection steps. In particular, five aptamers (sgc8, sgc3, sgd3, sgc4, and sgd2) were able to bind to target cells at high affinity, with K_D_ values ranging from 0.80 ± 0.09 nM (sgc8) to 26.6 ± 2.1 nM (sgc4), and other ALL cell lines, but not to cultured B cells and acute myeloid leukemia (AML) cells [48]. In addition, these aptamers could discriminate different leukemia cells (T-ALL, B-ALL, AML) in clinical samples, thus detecting subtle molecular differences among individual samples from leukemia patients in the same category [49]. Post-SELEX target identification, based on aptamer-mediated affinity purification and mass spectrometry, allowed the match of sgc8 aptamer with its target, the transmembrane receptor protein tyrosine kinase 7 (PTK7) [50]. Subsequently, as discussed below, this aptamer was applied as a new therapeutic tool for haemato-oncological malignancies [51]. By using a similar strategy, Tan’s group also identified the immunoglobin heavy mu chain as the target for TD05 aptamer, which was selected by using the Burkitt’s lymphoma cell line Ramos as the target [52]. Both of these studies demonstrate that this two-step strategy, the development of high-quality aptamer probes and the identification of their target proteins, is a powerful approach for biomarker discovery.

Through the use of several protocols, essentially based on altering positive selection steps on the chosen target cells and counter-selection steps on off-target cells, several aptamers have been to date developed with binding to unknown but unique characteristic surface proteins of target cells. For instance, aptamers have been generated that are able to discriminate high-metastatic from low-metastatic cancer cells of different tumor types, including colorectal carcinoma [53], breast cancer [54,55,56], osteosarcoma [57], prostate cancer [58,59] hepatocellular carcinoma [60,61,62] and colon cancer [63,64]. The generation of the above aptamer probes specifically targeting metastatic cancer cells provides a significant tool for diagnosis and treatment of the metastatic disease. For one of them, the investigation has already progressed to the target identification. Indeed, the RNA Apt63 aptamer, generated by a cell-SELEX approach for differential binding to prostate cancer cell lines with high vs. low metastatic potential, when used for aptamer-based affinity purification combined with mass spectrometry, matched to the plasma membrane ATP synthase beta subunit (ecto-ATP5B). Testing of the aptamer in vitro, as well as in xenograft models and human samples, proved this protein is a new marker for predicting and treating metastatic breast and prostate cancers [59]. Moreover, applied to cancer cells belonging to different tumor types (pancreatic cancer PANC-1 cells vs. hepatocarcinoma Huh7 cells) a blind cell-SELEX approach raised an aptamer able to regulate epithelial–mesenchymal transition (EMT) and inhibit metastasis in pancreatic cancer by binding to cell-surface vimentin, as revealed by post-SELEX liquid chromatography tandem mass spectrometry analyses [65].

It is known that cancer stem cells (CSCs) represent a small fraction of cells within a tumor mass exhibiting self-renewal and tumor-initiating capabilities, which contribute to recurrence, metastasis and therapeutic resistance. Unfortunately, specific biomarkers for CSCs are lacking, thus it remains extremely hard to eradicate them by effective therapeutic strategies [66]. Notably, differential cell-SELEX has been applied to identify new markers of CSCs. Some groups succeeded in generating aptamers able to differentiate glioma stem cells from differentiated cells [67,68], highlighting the potential of aptamers to target a molecular signature of CSCs for therapeutic applications. Similarly, aptamers have been generated to bind stemness-enriched cells in colorectal [69], pancreatic [70] and prostate [71] cancers. The identification of the molecular targets of these aptamers may reveal new CSC biomarkers.

Further, to address the issue of resistance to therapy, cell-SELEX has been applied to discriminate drug-resistant cancer cells from sensitive counterparts. Recently, by using vemurafenib-resistant melanoma cells as a target of the selection and sensitive cells in counter-selection steps, Li et al. identified an aptamer specifically binding to CD63 on the surface of cancer cells [72], thus opening the possibility to interfere with the TIMP-1/CD63 interaction at the cell surface, which recently emerged as a driver of malignant progression in melanoma and other human cancers [73]. Furthermore, by a SELEX approach using taxol-resistant colon cancer cells in the positive selection and parental cells in counter-selection steps, Zhang et al. identified a DNA aptamer binding to human TfR at affinity comparable to that of human Tf. Importantly, they proved the ability of the aptamer to cross the intestinal epithelium barrier through TfR-mediated transcytosis, indicating its potential as a carrier for active drug delivery [74].

In addition, we applied the differential cell-SELEX approach to different tumors, including NSCLC [75] and glioblastoma (GBM) [76], to generate aptamers able to target, within the same tumor type, cells characterized by a phenotype more aggressive than that of the cells used in the counter-selection, in terms of resistance to chemotherapy and tumorigenicity. Combining different post-SELEX biochemical approaches, we were able to identify EGFR and platelet-derived growth factor receptor beta (PDGFRβ) as the molecular targets of CL4 and Gint4.T aptamers, respectively, with the first coming from the selection on NSCLC cells [75] and the second from that on GBM cells [76]. Both 2′-fluoro-pyrimidine (2′-F-Py) containing RNA aptamers were subsequently validated as ligands and inhibitors of their proper receptor targets, not only when applied to cancer cells used for their selection but also in different tumor types, thus contributing to insight on the oncogenic role of these two RTKs, depending on the specific tumor expressing them. For instance, the anti-EGFR CL4 aptamer exerts a strong apoptotic effect on human NSCLC [75], blocks the invasiveness of GBM cells expressing either the EGFRwt or EGFRvIII mutant [76,77] and prevents the EGFR/integrin αvβ3 interaction on the surface of TNBC cells with a mesenchymal stem-like phenotype, which we first found to be required for vasculogenic mimicry and tumor growth of aggressive and poorly differentiated TNBC subtype [78,79].

Similarly, the Gint4.T aptamer acts as a neutralizing ligand for PDGFRβ in cell lines, primary cultures and xenografts models of GBM [76]. Further, it has been applied as a highly effective tool for imaging and suppression of TNBC lung metastases, thus indicating PDGFRβ as a reliable biomarker of a subgroup of TNBCs with invasive and stemlike phenotype [79,80]. More recently, Gint4.T has been proven to potentiate the efficacy of immunotherapy with anti-programmed death-ligand 1 (PD-L1) mAb in the inhibition of tumor growth and metastasis in a syngeneic TNBC mouse model [81].

Moreover, several cell-SELEX protocols have been applied to cancer cell lines by using normal cells for the counter-selection with the aim to identify novel cancer biomarkers for improving early diagnosis and therapy. For instance, 2′-F-Py RNA aptamers were selected on two different pancreatic cancer cell lines able to bind target cells and discriminate them from normal pancreatic ductal cells. Interestingly, aptamer-based target pull-down experiments on cell membrane lysates, combined with a genome-wide microarray analysis in cells targeted or not by the aptamer, identified the oncofetal protein alkaline phosphatase placental-like 2 (ALPPL-2) as the target of one of the selected aptamers and attributed a novel function to this protein as promoter of pancreatic cancer cell growth and invasion [15]. By a similar approach, four aptamers were identified that differentiate nasopharyngeal cancer cells from nonmalignant nasopharyngeal cells, the cell-surface receptor CD109 identified as the target of one of them [82]. The overexpression of CD109 in many human cancers and its association with metastasis and chemoresistance makes it an attractive target for diagnosis and therapy [83].

Moreover, in order to generate aptamers against tumor cells in more physiologically conditions, 3D cell-SELEX protocols have been applied to spheroids of prostate [84] and breast [85] cancer cells by using nontumor cells for the negative selection.

## 4. Unravel Cancer Heterogenicity by Cell-SELEX

One of the major factors limiting the number of effective targeted therapies is represented by cancer heterogeneity. Indeed, cancer is a dynamic disease and various cell subpopulations develop during its course, which are characterized by unique genotypes and phenotypes correlating to different biological behaviors and sensitivity to treatments. Distinct molecular signatures can be found among different tumors or inside the same one (inter- and intratumor, respectively). At the basis of the intratumor heterogeneity, there is a continuous process of evolution that involves all tumor cells, not only in time but also in space. For instance, Zhao et al. observed that the location of genetically distinct subclones in a tumor reveals their evolution, with the most aggressive and prone to metastasis located in the center of the tumor [86].

Phenotypic heterogeneity in different cell populations contributes to cancer drug resistance and hampers treatment outcome [87], but sufficient characteristics have not yet been determined to allow discrimination of the subpopulations present in a tumor. Cell-SELEX is an important approach for unraveling differently expressed cell-surface proteins of heterogeneous cancers such as GBM, breast and pancreatic cancers, which still lack personalized treatment protocols.

### 4.1. GBM

GBMs, classified as grade 4 astrocytomas, are the most frequent and aggressive malignant tumors occurring in the human brain and, according to gene expression, they were classified by the TCGA consortium into four molecular subtypes (proneural, mesenchymal, neuronal and classical) associated with different prognosis [88]. One of the main reasons for GBM aggressiveness is its intrinsic intratumor heterogeneity; indeed, it is composed of tumor stem cells, differentiated tumor cells, cells from the blood vessels, and inflammatory cells. The existence of these different subpopulations and their interactions with the TME components contributes to resistance to conventional treatments [89].

In order to create new opportunities for better treatment options, Lin et al. [90] applied a cell-SELEX protocol to human glioma SHG44 cells, a grade 3 human anaplastic astrocytoma, by using human astrocyte SVGp12 cells, for counter-selection. Aptamer S6-1b, obtained by truncating the long version coming from the selection, showed excellent specificity, discriminating target cells from SVGp12 cells and other cancer cell lines, including GBM U87, T98G and U251 cells. Cy5-labeled aptamers, intravenously injected in mice bearing subcutaneous SHG44-derived tumors, rapidly accumulated in the tumor and persisted for over 4 h, resulting in useful tools for noninvasive imaging of the tumor. Aptamer-mediated affinity purification of cell membrane proteins identified fibronectin, an extracellular matrix protein overexpressed in glioma, as the potential target of S6-1b.

Further, using SELEX technology on T98G target cells and SVGp12 off-target cells, Wu et al. selected two aptamers, WYZ-41a and WYZ-50a, able to bind to T98G but not to either U87 and U251 GBM cell lines [91] consistently, with major differences at the cell surface among them [92]. The aptamers were stable for 2 h and retained their binding capability in cerebral spinal fluid, thus indicating a great potential in using these aptamers for biomedical applications.

In addition, among a group of DNA aptamers selected on a cell line (K308) derived from gliosarcoma, a variant of GBM, WQY-9 aptamer was capable of recognizing gliosarcoma cells and gliosarcoma tissues, differentiating gliosarcoma from GBM [93].

### 4.2. Breast Cancer

Luminal A, Luminal B, HER-2 enriched and TNBC represent the four main subtypes of breast cancer [94]. They are characterized by different behavior and therapeutic sensitivity. Hence, dissecting the heterogeneity of its subpopulations is also necessary for the management of breast cancer. A differential cell-SELEX was applied to HER2-overexpressing breast cancer SK-BR-3 cells by using a pool of three counter-selections against Luminal A MCF-7 cells, breast normal MCF-10A cells and TNBC MDA-MB-231 cells. The length-optimized sk6Ea aptamer, resulting in the best binding candidate, was able to bind specifically to the target cells without recognizing the three counter-selection cell lines both in vitro and in vivo, upon intravenous injection in subcutaneous tumor-bearing mice. Moreover, sk6Ea when cemented on breast cancer tissue sections, efficiently discriminated the three different breast cancer subtypes [95].

TNBC, defined by the lack of ER, progesterone receptor (PR), HER-2 expression, is significantly more aggressive than other breast cancers, diagnosed at a later stage and more likely to develop recurrence. It is highly heterogeneous and the presence of different subtypes, related to different cancer features, makes it difficult to establish an appropriate therapy [94]. Lehman et al. classified TNBC into four subtypes: Basal-like 1, Basal-like 2, mesenchymal and luminal androgen receptor [96,97]. Recently, we applied a cell-SELEX method for the specific recognition of MDA-MB-231 cells, a highly metastatic human cell line representing an established model for the aggressive and undifferentiated mesenchymal TNBC subtype [98]. At each round, the positive selection step on TNBC cells was preceded by counter-selection against non-TNBC breast cancer BT-474 cells, which express high level of ER, PR, and HER-2, and A431 epidermoid cancer cells. By using high-throughput NGS and bioinformatics, we identified a panel of six 2′-F-Py RNA aptamers able to bind to human TNBC cell lines (MDA-MB-231, BT-549, MDA-MB-436, DU4475, MDA-MB-468) covering different TNBC subtypes, and to distinguish them from both normal cells and non-TNBC breast cancer cell lines representative of luminal A (MCF-7 and T47D) and HER2-positive (SK-BR-3) molecular categories. Further, these aptamers can be used for staining of histological tissues to differentiate TNBC human samples, showing a distinct pattern of binding on different tumors [98]. In addition to their binding capability, the aptamers actively internalize into target cells and inhibit the mammosphere-forming ability of TNBC cell lines. Therefore, we anticipate that the identification of their targets will help to expand the limited repertoire of actionable TNBC biomarkers.

### 4.3. Pancreatic Cancer

Pancreatic cancer is classified as basal, with the worst prognosis and classical clinical subtypes. More than 70% of patients are unresponsive to the current therapies, and despite many efforts made in past years there are no improvements, principally due to its great heterogeneity [99]. To determine if there were molecular markers differently expressed on pancreatic cancer, Yoon et al. investigated the expression of seven different markers (five EMT markers, one proliferation marker, and one leukocyte antigen marker) on human cancer tissues by using a multiplexed tissue imaging mass cytometer platform. They observed a substantial heterogeneity both intratumor and in tumors of the same grade [100]. Nevertheless, a cell-SELEX protocol using pancreatic cancer PANC-1 cells for the positive selection and hepatocellular carcinoma cells for counter-selection led to the identification of mitochondrial heat shock protein 70 or mortalin, as a potential biomarker of pancreatic cancer [101]. This protein is expressed on the surface of cells belonging to some human cancers, such as colorectal, neuronal and pancreatic, but not to normal tissues.

### 4.4. TME

The role of stroma microenvironment in both solid tumors as well as malignant haematological disease is well-accepted. Based on the high level of heterogeneity in the nature of mutations present between patients and even within patients during the different stages of their disease as well as the clear role of tumor/stroma interactions in chemoresistance, targeting interactions between tumor cells and their stroma provide new therapeutic approaches [102]. Because of the role of tumor-associated macrophages (TAMs) in supporting tumor progression and dissemination and their linking with worse clinical outcome and resistance to conventional therapies, many efforts are focused on the development of TAMs-targeted strategies for cancer treatment. To this aim, Sylvestre et al. aimed to identify by cell-SELEX an aptamer able to target this cell population, discriminating it from resident (M0-like) or tumoricidal (M1-like) macrophages [103]. Despite the use of negative selection screens on monocytes and M0-like macrophages, and positive selection on M2-like macrophages, resembling TAMs, they selected an aptamer, named A2, that binds to both human M0- and M2-like macrophages and monocytes, probably due to overlapping receptor expression between M0- and M2-like macrophages. Conversely, A2 does not recognize M1-like macrophages or other leukocyte populations. Because of its binding and internalization into CD14+ but not CD16+ monocytes, which is also observed in vivo, A2 holds great potential for drug delivery approaches targeting monocytes. Based on the binding behavior of A2 to monocytes, they hypothesized and then confirmed that the aptamer binds to the CD14 receptor, which is expressed by both monocytes and macrophages.

## 5. Aptamer-Based Targeted Conjugates for Therapeutic Applications

The list of aptamers that target known cell-surface proteins of cancer cells, generated by either protein-SELEX or cell-SELEX or their combination, which are applied for targeted cancer therapy, is growing rapidly [104]. Moreover, some aptamers generated by cell-SELEX that bind at high selectivity to a defined cancer cell type, without the prior knowledge of the target, have been successfully functionalized with secondary agents for targeted cancer therapy, even though their molecular target has not yet been identified [60,105]. For example, the DNA aptamer GMT8, isolated by Gao et al. by cell-SELEX against U87 human GBM cells [106], has been extensively used as an efficient ligand to improve drug delivery to GBM even if its target is still unknown [106,107]. Similarly, the E3 aptamer, which was generated by a SELEX approach designed to obtain aptamers able to internalize into prostate cancer cells but not in normal prostate epithelial cells, when chemically conjugated to monomethyl auristatin F, efficiently targeted and inhibited prostate cancer growth in mice [108]. Further, the E3-based conjugates targeted and killed a wide range of human cancer cell types, thus revealing the aptamer, whose exact cellular target is still under investigation, as a universal platform for cancer targeting and therapy [109].

Taking as examples those aptamers that have been matched with their proper cell-surface protein targets, we highlight the great potential of aptamers to drive therapeutics specifically to cancer cells expressing those proteins (Table 1). The high versatility of aptamers for chemical modification indeed allows conjugating them to other aptamers, antibodies, drugs or drug-loaded nanoplatforms, thus increasing the spectrum of aptamer-based applications for cancer treatment. Sophisticated approaches have been performed to improve the active delivery of a drug as in the case of aptamers immobilization on the surface of nanomotors that allowed faster diffusion of the drugs into the cells [110].

### 5.1. CL4, Gint4.T and GL21.T Aptamers

Due to the high specificity and affinity of the aforementioned 2′-F-Py RNA CL4 aptamer for the EGFR [75], different kinds of conjugates have been generated and validated in preclinical cancer models (Figure 3).

Shu et al. constructed a trifunctional RNA nanoplatform containing the three-way junction (3WJ) core of the bacteriophage phi29 packaging RNA, incorporating the CL4 aptamer and anti-miRNA-21 as the tumor ligand and therapeutic, respectively (Figure 3A). Importantly, each component in the resulting RNA nanovector retained its proper folding and functionality, as well as sufficient stability due to the presence of 2′-F nucleotides, resulting in complete tumor regression when injected intravenously into mice bearing orthotopic TNBC xenografts [111]. A similar approach, based on the use of the 3WJ-EGFR CL4 aptamer, was used to deliver small interfering RNA (siRNA) against XBP1, a protein involved in breast cancer chemoresistance, to TNBCs, thereby sensitizing them to chemotherapy (Figure 3B). 2′-F-Pys were used to confer nuclease stability to the RNA nanoparticles (NPs) while keeping the advantages of the RNA, i.e., easy synthesis and minimal adverse immune response [112].

Apart from the use of CL4 to deliver therapeutic siRNAs and miRNAs to EGFR-positive tumors, Guo et al. generated four-way RNA junction (RNA-4WJ) NPs, redesigned from pRNA-3WJ motif, for solubilizing and high-yield loading of paclitaxel (Figure 3C). These nanostructures were then used to treat mice bearing MDA-MB-231 TNBC xenografts, thus overcoming the poor bioavailability of the drug [113].

More recently, we synthesized cisplatin-loaded poly(lactic-co-glycolic)-block-poly ethylene glycol (PLGA-b-PEG)-based polymeric nanoparticles (PNPs), capable of specifically guiding the chemotherapeutic agent to TNBC xenografts in mice, thanks to the EGFR aptamer that decorates the nanoparticle’s surface (Figure 3D). Notably, the CL4-equipped nanovectors showed significantly higher tumor targeting and anticancer activity than naked cisplatin and untargeted nanoparticles, and were also well tolerated with no signs of systemic toxicity [114].

Moreover, the finding that the combined treatment of the EGFR aptamer with anti-HER2 antibodies or the immune-checkpoint anti-PD-L1 and anti-CTLA-4 mAbs modulators efficiently inhibits the growth of several cancer cell types led us to the construction of bispecific aptamer-antibody conjugates having more potent cytotoxic effects than single agents [41,42] (Figure 3E). In these conjugates, the addition of an aromatic aldehyde functional group to the 5′ terminus of the aptamer allowed covalent linking to the antibody modified with HyNic functional groups; however, different chemistries need to be explored in order to increase the final yield of the construct for in vivo validation. The use of aptamer-based immunoconjugates may be an attractive platform for effective cancer therapy due to increased specificity for tumor cells, efficient activation of T cells against cancer cells and combined beneficial properties of both the aptamers and antibodies, such as high tissue penetration and long half-life in circulation, respectively [19].

In addition, the aptamers selected by differential cell-SELEX and targeting Axl [115] and PDGFRβ [76], in addition to their use as stand-alone therapeutics, have been applied as ligands for cancer-targeted delivery approaches. Specifically, the first one has been deeply investigated as a component of chimeric molecules for delivering therapeutic miRNAs to Axl-positive tumors [116] and the second one as a targeting agent for effective GBM therapy, thanks to its dual function for both BBB crossing and cancer targeting [117]. Indeed, the PDGFRβ Gint4.T aptamer successfully delivered PLGA-b-PEG PNPs loaded with the low-water-soluble NVP-BEZ235 drug to GBM intracranially implanted in mice [117]. More recently, Gint4.T was assembled into tetrahedral DNA nanostructures, alone [118] or in combination with the GMT8 aptamer [107], to deliver doxorubicin or paclitaxel to GBM cells, respectively, by crossing in vitro models of BBB.

### 5.2. Sgc8c Aptamer

Another aptamer largely employed as delivery agent is sgc8, which targets the leukemia biomarker PTK7 [50] (Figure 4).

Researchers have explored delivering the anthracycline Daunorubicin (Dau) agent to leukemia cells upon conjugation to different nanoplatforms. Taghdisi et al. adsorbed both sgc8 and Dau onto single-walled carbon nanotubes (SWNTs), molecular-scale tubes of graphitic carbon, through a stable noncovalent π electron interaction (Figure 4A). In such a complex, the aptamer increases SWNTs’ solubility and prevents their aggregation. On the other hand, SWNTs protect aptamers from nuclease degradation and extend their half-life. The resulting complex selectively internalized into the target human ALL Molt-4 T cells, where the chemotherapeutic agent was released in a pH-dependent manner [119]. Then, in order to explore more safe platforms, the same group used gold nanoparticles (AuNPs) that were Dau-loaded and conjugated on their surface with sgc8 aptamer alone [120] (Figure 4B) or in combination with the anti-nucleolin AS1411 aptamer to increase the cytotoxic activity of the complex on Molt-4 cells [121] (Figure 4C). Sgc8c-conjugated AuNPs were also used to efficiently deliver Dox to target cancer cells [122] (Figure 4D).

The AuNPs can be also conjugated, for combination therapy, to another sgc8 ap-tamer–platform: nanotrains (NTrs). This platform consisted of multiple repetitive “boxcars” that were self-assembled from two hairpin DNA monomers (M1 and M2) and the sgc8 aptamer modified on the 5′-end with a DNA trigger. Alternative to the AuNPs, which were attached to the thiol groups modified on the 5′-ends of M1 and M2, the boxcars were loaded with several widely used anthracycline anticancer drugs, including Dox, Dau and epirubicin (EPR) (Figure 4E). Aptamer-NTrs (aptNTrs) exhibit various positive features, including high payload capacity and specific targeting for cancer therapy. The high selectivity of sgc8-NTrs for delivery of Dox, intercalated into the double strand, to ALL has been shown both in vitro and in vivo. Mice bearing ALL treated with aptNTrs showed similar antitumor efficacy compared with free dox but reduced side effects [123].

Further, Tan’s group decorated the surface of gold nanorods (AuNRs) with the thiol-terminated sgc8 aptamer, which was linked at the 3′-end with chlorine e6 (Ce6), a highly effective photosensitizer. The binding of the aptamer to its protein target on cancer cells caused the release of Ce6 away from the gold surface, thus producing singlet oxygens for photodynamic therapy upon light irradiation. Importantly, the final therapeutic effect was improved by further cell destruction due to the photothermal effect caused by the absorption of radiation by AuNRs [124] (Figure 4F). After that, different nanomaterials were equipped with Sgc8 aptamer to specifically deliver drugs to ALL cells and favor their cell uptake in a receptor-mediated manner, including black phosphorus nanosheets [125] (Figure 4G), mesoporous silica nanoparticles [126] (Figure 4H), and glutathione-responsive polymeric micelles [127] (Figure 4I).

### 5.3. GBI-10 Aptamer

Tenascin-C (TNC) is an extracellular matrix glycoprotein almost undetectable in adult tissues but highly expressed in the TME, which correlates with poor patient survival in several malignancies, including glioma. Different binding partners of TNC have been so far identified and, depending on the specific cancer cell type, the interaction of TNC with its receptor induces cell proliferation, adhesion, invasion, and angiogenesis and escape from immune surveillance [132]. To overcome some limitations of TNC antibody-based therapy, aptamers against TNC were developed by both a SELEX approach combining GBM cells and recombinant TNC protein [133]; thus, with the precise knowledge of the target, and a cell-SELEX against GBM U251 cells and post-SELEX target identification [128]. The DNA aptamer derived from the last approach, named GBI-10, was successfully used in several diagnostic applications [134] and, more recently, conjugated to nanoparticles to improve tumor penetration and drug release. For example, He et al. [129] designed a very smart nanovector for precision drug delivery to pancreatic cancer by using the negative charge of GBI-10 aptamer as camouflage for a cell-penetrating peptide (CPP) in order to reduce the untargeted systemic accumulation of the nanovector. Then, the disulfide-containing dimeric camptothecin prodrug was encapsulated into amphiphilic polypeptide copolymer that was functionalized with the aptamer-modified peptide. The binding of TNC to the aptamer detaches the aptamer from the complex, and the peptide is able to deliver the drug into cancer cells where the intracellular high redox potential allows the prodrug cleavage and subsequent induction of antitumor activity.

In another study, size-changeable nanoparticles were designed that maintained an initial large size for prolonged blood circulation and transformed into small ones within the tumor for better tumor penetration. Specifically, dandelion-like tailorable NPs were generated by using an acid-responsive linker to increase their size, which were loaded with the macrophage conditioning agent zoledronic acid and decorated with GBI-10 aptamer for tumor targeting. The exposure of resulting nanoparticles to the acidic TME caused de-crosslinking and reduction of NPs size with enhanced drug distribution in tumor site, as observed both in in vitro 3D TNBC spheroids and in mice bearing orthotopic TNBC 4T1 cell-derived tumors [130].

## 6. Conclusions

Even with incredible technological advancements, the search for new and unique biomarkers in cancer cells is still “a hit and miss” process. Furthermore, the techniques required for this, including data mining, protein identification, validation, development of therapeutic modalities, etc., add to the lengthy process as well as costs of development if successful. Aptamers were first described in 1990, and since then they have found several niche areas where they demonstrate superiority over similar technologies (i.e., mAbs). Their in vitro generation has led to a multitude of aptamers selected for known protein targets, as well as for small molecules all the way to whole organisms and has shown how versatile they are. In this review, we discussed one of these unique applications, which is the identification of novel targets while developing the therapeutic modality concurrently. This cell-SELEX can help with identifying biomarkers present on certain cancer types or in distinguishing metastatic cancer cells from ”normal” cancer cells, as well as drug-resistant cancer cells from drug-sensitive ones. We also presented information on new targets discovered by cell-SELEX that have been validated in vivo. These aptamers are undergoing further validation for therapeutic development, including attempts to improve their less-than-optimal pharmacokinetic and biodistribution profiles [19,135], though this may become easier with the advancements that have seen mRNA vaccines FDA-approved. While there are also developments in the monoclonal antibody field that allow for in vitro selection using antibody libraries, it is not just the ease of the SELEX process, but the stability and costs involved, that will see aptamers utilised for the identification of unique targets for cancer therapeutic development in the future.

## Figures and Tables

**Figure 1 pharmaceutics-14-00028-f001:**
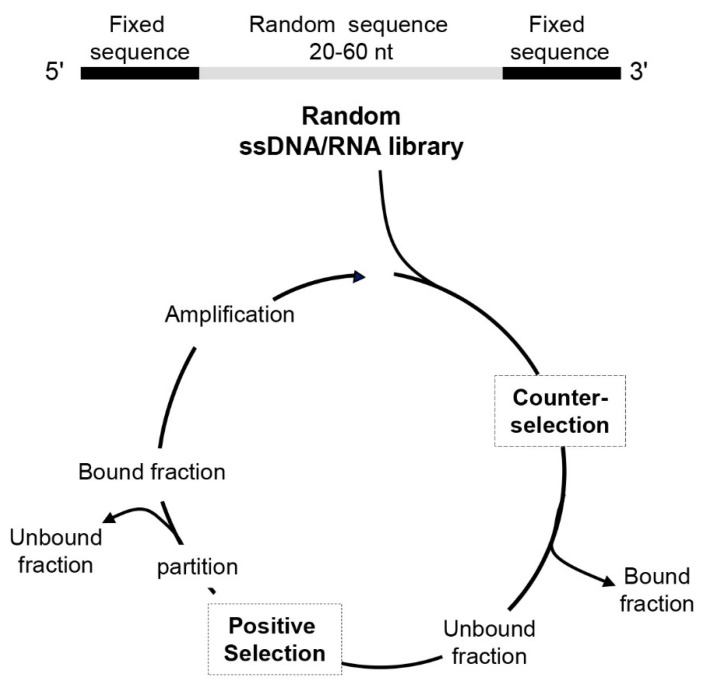
Schematic representation of the Systematic Evolution of Ligands by Exponential Enrichment (SELEX) technology. The starting point is the generation of a library of single-stranded DNA (ssDNA) or RNA that contains a randomized region flanked by two fixed primer-binding sequences on both ends to allow enzymatic amplification and in vitro transcription (in the case of the RNA library). The SELEX method consists of iterative steps of incubation, partition, recovery and amplification (see text for details).

**Figure 2 pharmaceutics-14-00028-f002:**
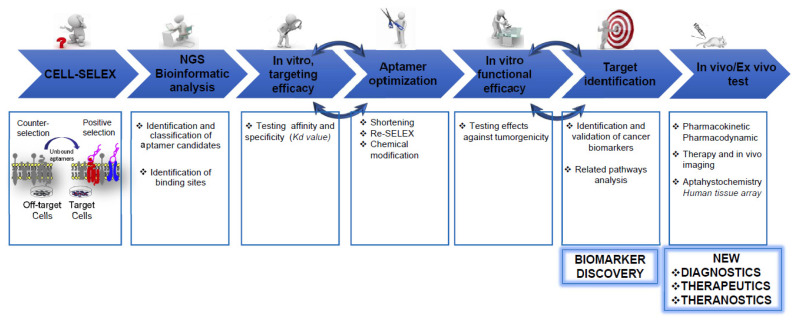
Aptamer discovery and development process. The image shows the sequential steps from the generation of aptamers by SELEX to their preclinical development as cancer-targeted therapeutics, imaging agents for diagnostics and theranostic molecules. Briefly, cancer cell-specific aptamers are generated by cell-SELEX against a specific cell type, without the prior knowledge of the molecular target. High affinity ligands are identified by classical cloning and/or high-throughput sequencing and bioinformatic analysis. The best aptamer candidates are tested in vitro for their affinity and specificity, optimized to increase their binding efficacy and tested in vitro/in vivo as cancer-targeted agents. The best binding aptamers are used as bait to identify their cell-surface targets, thus leading to discovery of novel biomarkers.

**Figure 3 pharmaceutics-14-00028-f003:**
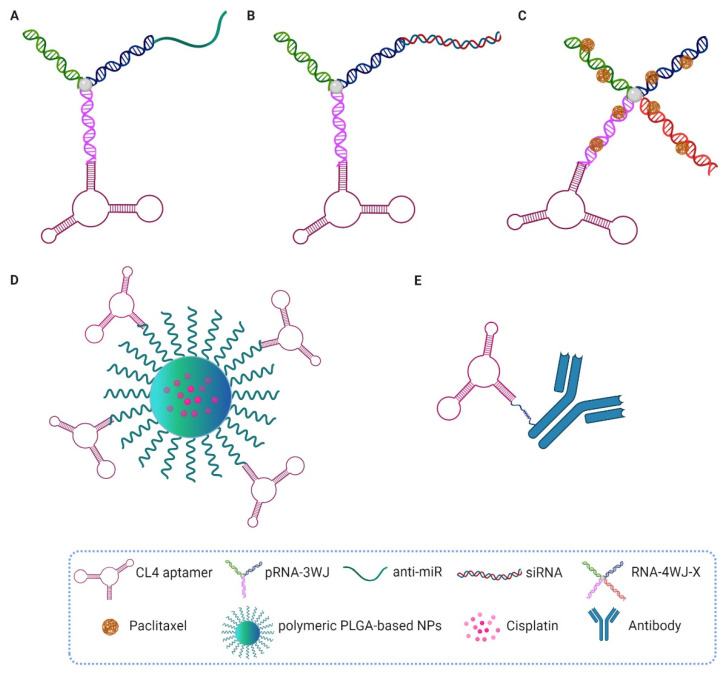
Schematic representation of the anti-EGFR CL4 aptamer-based conjugates. (**A**) 3WJ-CL4/anti-miRNA-21 nanoparticles. (**B**) 3WJ-CL4/siRNA-XBP1 nanoparticles. (**C**) 4WJ-CL4/paclitaxel nanoparticles. (**D**) CL4-PNPs/cisplatin. (**E**) CL4-antibody (see text for details). This figure was created by BioRender.com (accessed date 12 October 2021).

**Figure 4 pharmaceutics-14-00028-f004:**
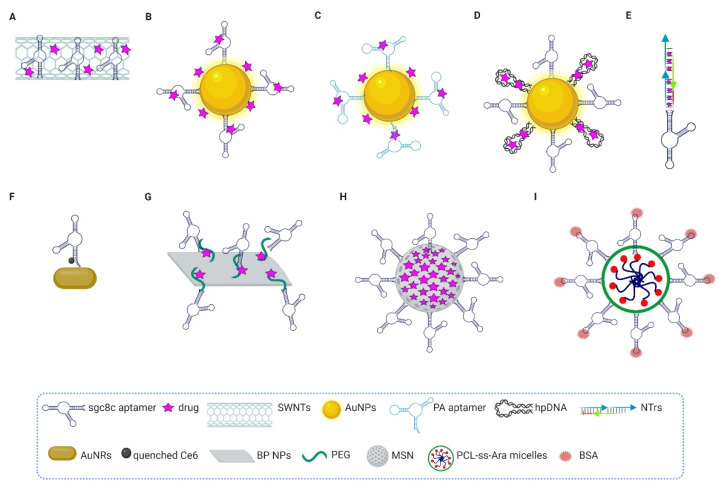
Schematic representation of the anti-PTK7 sgc8c aptamer-based conjugates. (**A**) sgc8 SWNTs/Dau. The sgc8 aptamer and Dau were loaded onto SWNTs through a π electron interaction. (**B**) sgc8-AuNPs/Dau. Sgc8 aptamer was assembled onto the surface of AuNPs and Dau was adsorbed on the surface of AuNPs by electrostatic interaction; a small amount of drug binds to sgc8, because of intercalating characteristic of Dau. (**C**) PA-AuNPs/Dau. A modified polyvalent aptamer and Dau were loaded onto the surface of AuNPs, as described in (**B**). (**D**) sgc8-hpDNA-AuNPs/ doxorubicin (Dox). Sgc8 and a d(CGATCG)-rich hpDNA, in which Dox is intercalated, were assembled onto the surface of AuNPs through standard gold-thiol chemistry. (**E**) sgc8-tethered DNA nanotrains (aptNTrs)/drug. AptNTrs obtained by self-assembly of the sgc8, modified on the 5′-end with a DNA trigger (red line), and two hairpin monomers (M1 and M2, blue and green arrows, respectively), and loaded with anthracycline anticancer drugs (Dox, Dau or epirubicin (EPR)). (**F**) sgc8-Ce6/AuNRs. The photosensitizer Ce6 was connected to the 3′-end of sgc8, containing a sulfhydryl group at the 5′-end for attachment to AuNRs. (**G**) sgc8-BP NS-PEG/Dox. Sgc8 was covalently conjugated onto the surface of BP NS, modified with PEG and loaded with Dox through electrostatic adsorption. (**H**) sgc8-MSN/Dox. Dox was encapsulated in MSN, which was decorated with carboxyl-modified sgc8. (**I**) sgc8-BSA/PCL-ss-Ara. The sgc8-bovine serum albumin (sgc8-BSA) decorated the surface of glutathione-responsive polymeric micelles formed though self-assembly of the prodrug, synthesized via covalent bond formation between amphiphilic copolymers of acryloyl chloride-terminal PCL-ss-PCL and Ara-C (4-NH2). This figure was created by BioRender.com (accessed date 12 October 2021).

**Table 1 pharmaceutics-14-00028-t001:** Aptamer-based conjugates for targeted drug delivery in cancer.

Aptamer	Composition	Size (nt)	Cell Type for the Selection	Post-SELEX Identified Target	Kd (nM)	Aptamer-Based Targeted Conjugates for Therapeutic Applications
CL4	2′-F-Py-RNA	39	NSCLC	EGFR [75]	10	⮚ 3WJ-CL4/anti-miRNA-21 [111] ⮚ 3WJ-CL4/siRNA-XBP1 [112] ⮚ 4WJ-CL4/paclitaxel [113] ⮚ CL4-PNPs/cisplatin [114] ⮚ CL4-antibody [41,42]
GL21.T	2′-F-Py-RNA	34	GBM	Axl [115]	13	⮚ GL21.T-miRNAs [116]
Gint4.T	2′-F-Py-RNA	33	GBM	PDGFRβ [76]	9.6	⮚ Gint4.T-PNPs/NVP-BEZ235 [117] ⮚ Gint4.T-TDN/doxorubicin [118] ⮚ Gint4.T-tFNA-GMT8/paclitaxel [107]
sgc8c	DNA	41	T-cell ALL	PTK7 [50]	0.78	⮚ sgc8- SWNTs/daunorubicin [119] ⮚ sgc8-AuNPs/daunorubicin [120] ⮚ PA-AuNPs/daunorubicin [121] ⮚ sgc8-hpDNA-AuNPs/doxorubicin [122] ⮚ sgc8-NTRs/drug or AuNPs [123] ⮚ sgc8-Ce6-AuNRs [124] ⮚ sgc8-BP NS-PEG/doxorubicin [125] ⮚ sgc8-MSN/doxorubicin [126] ⮚ sgc8-BSA/PCL-ss-Ara prodrug [127]
GBI-10	DNA	69	GBM	Tenascin C [128]	150	⮚ GBI-10-CPP/camptothecin prodrug [129] ⮚ GBI-10-DGLs/zoledronic acid [130]
SQ-2	2′-F-Py-RNA	25	Pancreatic cancer	ALPPL-2 [15]	20	⮚ SQ-2/5-fluoro-2′-deoxyuridine [131]

AuNPs: gold nanoparticles; AuNRs: gold nanorods; BP NS: black phosphorus nanosheets; BSA: bovine serum albumin; Ce6: chlorin e6; CPP: cell-penetrating peptide; DGLs: dendrigraft poly-*L*-lysines; hpDNA: hairpin DNA; MSN: mesoporous silica nanoparticles; miRNA: microRNA; NTRs: nanotrains; PA: polyvalent aptamers (sgc8 and anti-nucleolin AS1411 aptamers); PEG: polyethylene glycol; PNPs: polymeric NPs; siRNA: small interfering RNA; SWNTs: single-walled carbon nanotubes; TDN: tetrahedral DNA nanostructures; tFNA: tetrahedral framework nucleic acid; 3WJ: three-way junction.

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
