# Peer review of "Profiling Cancer Cells by Cell-SELEX: Use of Aptamers for Discovery of Actionable Biomarkers and Therapeutic Applications Thereof"

_pharmaceutics, 2021, doi:10.3390/pharmaceutics14010028_

Round 1

Reviewer 1 Report

The review deals with cell-SELEX of aptamers specific to the surface determinants of tumor cells, their selection and potential applications for cell targeting and the discovery of novel biomarkers. A prominent accent is made on the own works of the authors which is quite fair due to their significant impact on the field.  The manuscript is well-structured and clearly written. I recommend it for publishing in its current form.

Reviewer 2 Report

The review submitted by Cerchia et al discuss about the contribution of cell-Selex for profiling cancer cells. It is an update of already published reviews on the same subject (eg Pang et al, Cancers, 2018).

It is well written and comprehensive. In my opinion, the author should also discuss about the size of the aptamers, indeed, it is not the same work to modify a 30-mer compared to a 100 mer. This information should be added in table 1

I also have a reservation about figure 1: it is to small and is unreadable especially the small scheme where the counter selection is explained (first frame).

I recommend publication after those modifications

Reviewer 3 Report

Shigdar et al review the potential and power of cell-SELEX used for generating cell specific or biomarker specific aptamers and therapeutic applications thereof with several drug delivery strategies. The review is well written, logically structured and was a pleasure to read. However, there are a few remarks that need to be addressed before publication.

Cell-SELEX has been reviewed before by e.g. Cibiel et al (Pharmaceuticals, 2011), Barman (RSC Advances, 2015), Chen et al (Int. J. Mol. Sci., 2016), Kaur (Biochem. et Biophys. Acta, 2018)  and Zhong et al (Anal. Biochem., 2020) and in a progress report by Bing etal (Advanced Biosystems, 2019), but I miss referencing to these overviews. Doing so, the authors include several more-or-less relevant individual Cell-SELEX papers they do not directly reference to themselves.

Also, very recently, Zhang and Shangguan (Int. J. Mol. Sci., 2021) published an interesting report in which they describe an aptamer against human transferrin that deserves some attention in the review.

Lastly, there are two groups that applied 3D cell-SELEX on spheroids of cancer cells, namely Souza et al (Exp. Cell Res., 2016) and Nelissen et al (Pharmaceuticals, 2021). Because the microenvironment in 3D models better mimic an in vivo situation and with that possible aptamer therapeutic efficacy, the authors should also spend a few lines on this topic in their review.

Minor / typos:

Line 140: decreased should be increased

Line 161: remove ‘process’ here?

Line 163: aptamer

Line 177: rephrase?

Line 242: replace ‘higher’ by ‘more’

Line 309: ‘removing useless nucleotides from’:  the authors could just use ‘truncating’ here.

Line 310: specificity

Line 313: rephrase ‘making it a good option’

Line 314: membrane

Line 341: makes it difficult

Line 528: aptamer

Reviewer 4 Report

The manuscript by Shigdar et al. (Pharmaceutics-1487624) provided an updated perspective of using aptamer for profiling cancers cells and therapeutic applications. The manuscript is very well written and organized. I feel that the manuscript may be published in its current form. Though, I have some minor suggestions that could improve the readability and value of the manuscript for readers that are not in the aptamer research field.

Suggested:

1) The authors have provided a vast amount of reference detailing the advancement in aptamer discovery and technology in cancer theranostics application, yet most if and not of the advancement is still in the pre-clinical stage. It would be better if the authors could provide a perspective in the conclusions to comment on the hurdle to advance aptamers into clinical-stage research. These factors could be but are not limited to PK/ PD factors. I feel the added comment would give readers a more balanced message on the advantage and the current limitation of aptamer technology faces.

2) If available, please add the KD values that correspond to the aptamers listed in Table 1. It would give readers a sense of how well these aptamers can bind at a glance.

Reviewer 5 Report

The review includes information about selection of aptamer by cell-SELEX that bind to various cancer markers at the surface of the tumor cells. It also includes examples of aptamer-nanostructure conjugates for targeted delivery of chemotherapeutics into the cells. The review is well written and can be useful for clinicians as well as for researchers wiring on the targeted drug delivery. It can be published after following improvements.

  1. Please include the scheme of cell-SELEX and explain all steps that resulting in selection of the aptamers. Explain how the initial library of DNA oligos is formed. Explain the principle of negative selection.
  2. Explain the principles of post SELEX modification and application of computer modeling for improvement of the aptamer binding properties.
  3. Include examples of aptamer sequences and explain their secondary and 3D structure. Discuss the formation of biding site for aptamer in a solution and the factor that affect this.
  4. Aptamers have been immobilized also on the surface of nanomotors that allowed faster diffusion of the drugs into the cells, or allowed the monitoring of the cancer markers for the purpose of the cancer diagnosis. See for example Beltrán-Gastélum et al. ChemPhysChem 2019, 20, 3177– 3180.
  5. In the citations at the text avoid authors initials. For example instead of Yoon S. et al. write Yoon et al.
  6. Figure 1 should be prepared in better resolution.
